behaviour, developmental biology

autism spectrum disorder, broad autism phenotype, facial morphology, masculinity, photogrammetry

**Author for correspondence:**
Diana Weiting Tan
e-mail: diana.tan@uwa.edu.au

# An investigation of a novel broad autism phenotype: increased facial masculinity among parents of children on the autism spectrum

Diana Weiting Tan[1,2], Syed Zulqarnain Gilani[3,4], Gail A. Alvares[2], Ajmal Mian[3], Andrew J. O. Whitehouse[2] and Murray T. Maybery[1]

[1]School of Psychological Science, The University of Western Australia, 35 Stirling Highway, Perth, WA 6009, Australia
[2]Telethon Kids Institute, [3]Centre of AI & ML, School of Sciences, and [4]Institute for Nutrition Research, Edith Cowan University, Perth, Australia

 DWT, 0000-0002-6394-8435; SZG, 0000-0002-7448-2327; GAA, 0000-0003-3351-5919; AM, 0000-0002-5206-3842; AJOW, 0000-0001-8722-1575; MTM, 0000-0002-3760-1382

The broad autism phenotype commonly refers to sub-clinical levels of autistic-like behaviour and cognition presented in biological relatives of autistic people. In a recent study, we reported findings suggesting that the broad autism phenotype may also be expressed in facial morphology, specifically increased facial masculinity. Increased facial masculinity has been reported among autistic children, as well as their non-autistic siblings. The present study builds on our previous findings by investigating the presence of increased facial masculinity among non-autistic parents of autistic children. Using a previously established method, a 'facial masculinity score' and several facial distances were calculated for each three-dimensional facial image of 192 parents of autistic children (58 males, 134 females) and 163 age-matched parents of non-autistic children (50 males, 113 females). While controlling for facial area and age, significantly higher masculinity scores and larger (more masculine) facial distances were observed in parents of autistic children relative to the comparison group, with effect sizes ranging from small to medium ($0.16 \leq d \leq .41$), regardless of sex. These findings add to an accumulating evidence base that the broad autism phenotype is expressed in physical characteristics and suggest that both maternal and paternal pathways are implicated in masculinized facial morphology.

## 1. Introduction

Autism spectrum disorder (hereafter 'autism') is a neurodevelopmental condition characterized by differences and difficulties in social communication, and the presence of special interests and repetitive behaviours [1]. Autism is a heritable condition, with the likelihood of diagnosis increasing as a function of genetic relatedness [2]. The 'broad autism phenotype' commonly refers to sub-clinical levels of autistic-like behaviour and cognition which have been observed in biological relatives of autistic people.[1] Compared to parents without a family history of autism, biological parents of autistic children have reported increased circumscribed interests [4], and poorer language and social communication skills [5]. Recent evidence has suggested that the broad autism phenotype may extend beyond behaviour and cognition, and may also express in physical forms, specifically, in facial morphology [6,7].

Earlier prevalence studies have suggested that males are two to three times more likely to receive an autism diagnosis than females [8,9]. The male preponderance in autism diagnosis has led to the development of the 'androgen

hypothesis', which proposes that autism is associated with an increased exposure to prenatal androgens, including testosterone [10,11]. Testosterone is a powerful masculinizing sex steroid crucial for male virilization during fetal development; hence, its production is much more pronounced in developing male fetuses than female fetuses [12]. Testosterone also crosses the blood–brain barrier and can influence fetal brain development during pregnancy [13].

Most previous studies investigating the relationship between prenatal testosterone and autism have focused on individuals recruited from the general population, partly due to restricted availability of biological specimens for the analyses of prenatal testosterone. Studies based on autistic-like traits in the general population have shown mixed findings, with some studies reporting a positive correlation between levels of prenatal testosterone and parent-reported autistic-like traits among non-autistic children [14,15], and other studies reporting null associations [16–18]. To date, only one study has analysed concentrations of prenatal testosterone among boys with a clinical diagnosis of autism [10]. By linking steroidal data derived from amniotic fluid samples biobanked in the Danish Historic Birth Cohort to records of boys diagnosed with autism archived in the Danish Psychiatric Central Register, Baron-Cohen et al. [10] found that a latent factor score comprising levels of prenatal testosterone and its precursor and derivative steroid compounds (e.g. androstenedione) was significantly elevated in autistic boys ($n = 128$) relative to non-autistic boys ($n = 217$). Data of autistic girls were not analysed due to low sample size ($n = 24$). In sum, the current evidence for the relationship between prenatal testosterone and autistic-like traits in the general population appears to be mixed, whereas there has only been one study conducted in the clinical population reporting a positive association.

Testosterone exposure is also connected to the development of secondary sex characteristics such as facial masculinity. In one of our previous studies, we found that higher concentrations of testosterone derived from umbilical cord blood collected at birth were associated with more masculinized facial features in adulthood, for both men ($n = 86$) and women ($n = 97$) [19]. Facial masculinity was determined using three-dimensional photogrammetry combined with a suite of gender classification and scoring algorithms including a gradient-based efficient feature selection (GEFS) algorithm [20] and a linear discriminant analysis (LDA) classifier [21]. In further research, we applied a similar methodological approach to investigate whether autistic children presented with greater facial masculinity compared to non-autistic children [22]. We found that three-dimensional images of autistic children (54 boys and 20 girls) showed more pronounced facial masculinity compared to their same-sex counterparts without autism (54 boys and 60 girls).

In a recent study, McKenna et al. [23] replicated our observations using two-dimensional images of 216 males and 129 females diagnosed with a neurodevelopmental condition (including 181 autistic people), and 165 males and 253 females without any such condition. McKenna et al. [23] reported increased facial masculinity among males and females with a neurodevelopmental condition compared to their same-sex counterparts without any condition. The authors further built on these findings by examining the association between facial masculinity and polygenic risk scores associated with sex-hormone binding globulin

(SHBG) derived from genomic analyses. SHBG is a steroid-binding protein that transports testosterone in blood and regulates steroidogenic activities [24]. When bound to SHBG, testosterone becomes biologically inactive. Thus, low concentrations of SHBG have been associated with heightened levels of biologically active testosterone [24]. McKenna et al. [23] found that lower SHBG polygenic risk scores significantly correlated with greater facial masculinity, suggesting that greater facial masculinity associated with neurodevelopmental conditions may be driven by excess biologically active testosterone.

More generally, facial structure is highly influenced by genetic factors [25] with biological relatives looking more alike than strangers without familial ties. From three-dimensional facial images of monozygotic and dizygotic twins, Djordjevic et al. [26] estimated that genetic factors explained more than 70% of the facial morphological variation. In terms of facial masculinity, several monozygotic–dizygotic twin studies reported that genetic factors accounted for approximately 40–50% of the variability in facial masculinity [27,28].

Given that both autistic-like traits and facial masculinity are heritable characteristics that are likely to be a result of intricate interactions between genetic and environmental factors [23,25], our research group went on to examine facial masculinity expressed in the three-dimensional images of non-autistic siblings of autistic children. Relative to non-autistic children without a family history of autism (69 boys and 60 girls), age- and sex-matched non-autistic siblings of autistic children (30 boys and 25 girls) presented with more masculinized facial structures [7].

## (a) Present study

In summary, we have presented evidence that early exposure to elevated testosterone concentrations is associated with more masculinized facial structures for both sexes in adults [19]. Consistent with the androgen hypothesis, we reported increased facial masculinity in three-dimensional images of autistic boys and girls; McKenna et al. [23] reported similar results using two-dimensional images of people with neurodevelopmental conditions including autism. Furthermore, we have found increased facial masculinity in the three-dimensional images of non-autistic biological siblings of autistic children [7], thereby providing evidence for a physical expression of the broad autism phenotype. Taken together, a logical next step in this line of research is to examine facial masculinity of non-autistic biological parents of autistic children. Such an investigation will advance knowledge in two significant ways. First, if parents of autistic children also demonstrate masculinized facial structures, this would add further evidence to the suggestion that facial masculinity is a physical expression of the broad autism phenotype that is likely to be influenced by genetic rather environmental factors. Second, it will also be important to understand whether facial masculinization is evident in fathers, mothers or both parents, thus establishing whether facial masculinity expressed in autistic children is associated with paternal or maternal lineages.

Accordingly, in the present study, we examined facial masculinity for non-autistic biological parents of autistic children relative to age-matched parents of non-autistic children, based on their three-dimensional facial images analysed

using a previously established gender classification and scoring algorithm [7,19,22]. We hypothesized that facial masculinity would be more pronounced among parents of autistic children compared to parents of non-autistic children.

## 2. Methods

### (a) Participants

The sample[2] included 58 non-autistic biological fathers (age: M = 43.4 years, s.d. = 7.40) and 134 non-autistic biological mothers (age: M = 41.9 years, s.d. = 5.93) of autistic children who were recruited from the Telethon Kids Institute in Perth, Western Australia. All parents reported having at least one child who had received a diagnosis of autism spectrum disorder based on the diagnostic and statistical manual of mental disorders 4th or 5th edition (DSM-IV/5) criteria [1]. A comparison group of 163 parents who reported no known family history of autism (50 fathers, age: M = 41.9 years, s.d. = 8.32; 113 mothers, age: M = 39.1 years, s.d. = 7.44) were recruited from the general population at several community events held in Perth, Western Australia. As facial structures are known to be influenced by ethnicity [29], the presence of syndromic disorders [30] or significant facial trauma, only participants of Caucasian descent without known syndromes or histories of facial trauma were included in this study.

### (b) Three-dimensional facial photography

Frontal three-dimensional images of participants were captured using a 3dMDface system (3dMD, Atlanta, GA, USA), which has been shown to generate highly precise and reliable images [31]. During photography, each participant sat in front of the 3dMDface system and was asked to maintain a neutral facial expression with their lips closed.

### (c) Gender classification and scoring algorithm

Machine learning classifiers are commonly used in autism research [7,22,23] and were also used in a related study that reported a link between early testosterone exposure and facial masculinity in adults [19]. Therefore, to allow for continuity and comparison with previous findings, we implemented a machine learning approach established in previous studies [7,21,22]. First, 21 facial landmarks were selected for automatic annotation in the first instance using dense correspondence. The landmarks on each image were then checked and manually corrected by D.W.T. Next, 26 Euclidean (i.e. a straight-line distance between any two given points) and 26 geodesic (i.e. a curved distance over a surface between any two given points) distances were measured between these landmarks (figure 1 and table 1). The use of both geodesic and Euclidean measurements available in three-dimensional images has been shown to significantly improve gender classification accuracy [21].

We then applied a GEFS algorithm [20] to evaluate all possible combinations of the 52 facial distances and select a set of distances that maximally contributed to the overall accuracy in the gender classification of the current sample. The selected facial distances were then used to train a gender classifier using a LDA with 10-fold cross-validation [21].

A facial masculinity score for each image was then computed based on the deviation in measurements for each individual face from the mean measurements for males and for females (see figure 2 for a fuller description of the calculation of masculinity scores). Facial masculinity scores could range from 0 (highly feminine) to 1 (highly masculine). Additionally, facial area for each image was calculated using the three-dimensional point cloud that defined each facial model and the triangular connection between these points [7,19,22].

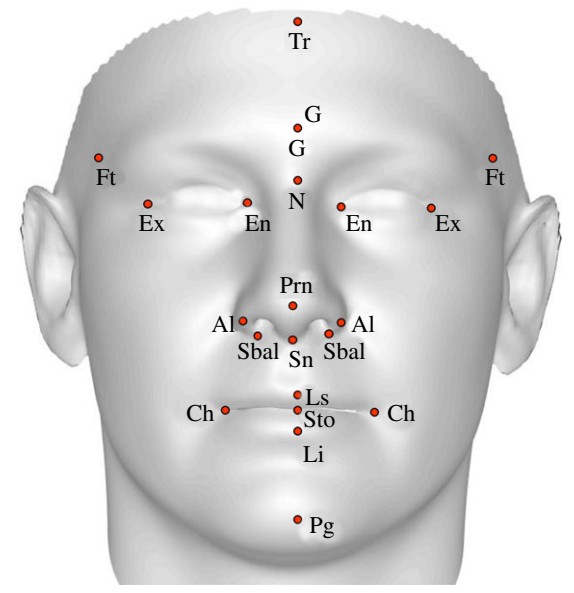

**Figure 1.** A composite facial image annotated with the 21 landmarks used in the current study. (Online version in colour.)

**Table 1.** Summary of facial landmarks and distances as defined by Farkas [32], which were measured in Euclidean and geodesic forms, and entered into the GEFS algorithm for feature selection.

| number | landmark | facial distance |
| --- | --- | --- |
| 1 | Ft-Ft | forehead width |
| 2 | Ex-Ex | outer canthal width |
| 3 | Ex-En (left) | eye fissure length (left) |
| 4 | Ex-En (right) | eye fissure length (right) |
| 5 | En-En | inter canthal width |
| 6 | Ex-N (left) | mid face width (left) |
| 7 | Ex-N (right) | mid face width (right) |
| 8 | En-N (left) | nasal root height (left) |
| 9 | En-N (right) | nasal root height (right) |
| 10 | Al-Al | nose width |
| 11 | Sbal-Sbal | alar-base width |
| 12 | Ch-Ch | mouth width |
| 13 | Ch-Pg (left) | mandible height (left) |
| 14 | Ch-Pg (right) | mandible height (right) |
| 15 | Ex-Ch (left) | upper cheek height (left) |
| 16 | Ex-Ch (right) | upper cheek height (right) |
| 17 | Tr-G | forehead height |
| 18 | N-Prn | nasal bridge length |
| 19 | N-Sn | nose height |
| 20 | N-Sto | upper facial height |
| 21 | Sn-Prn | nasal tip protrusion |
| 22 | Sn-Sto | upper lip height |
| 23 | Sn-Ls | philtrum length |
| 24 | Ls-Sto | upper vermillion height |
| 25 | Sto-Li | lower vermillion height |
| 26 | Sto-Pg | mandible height |

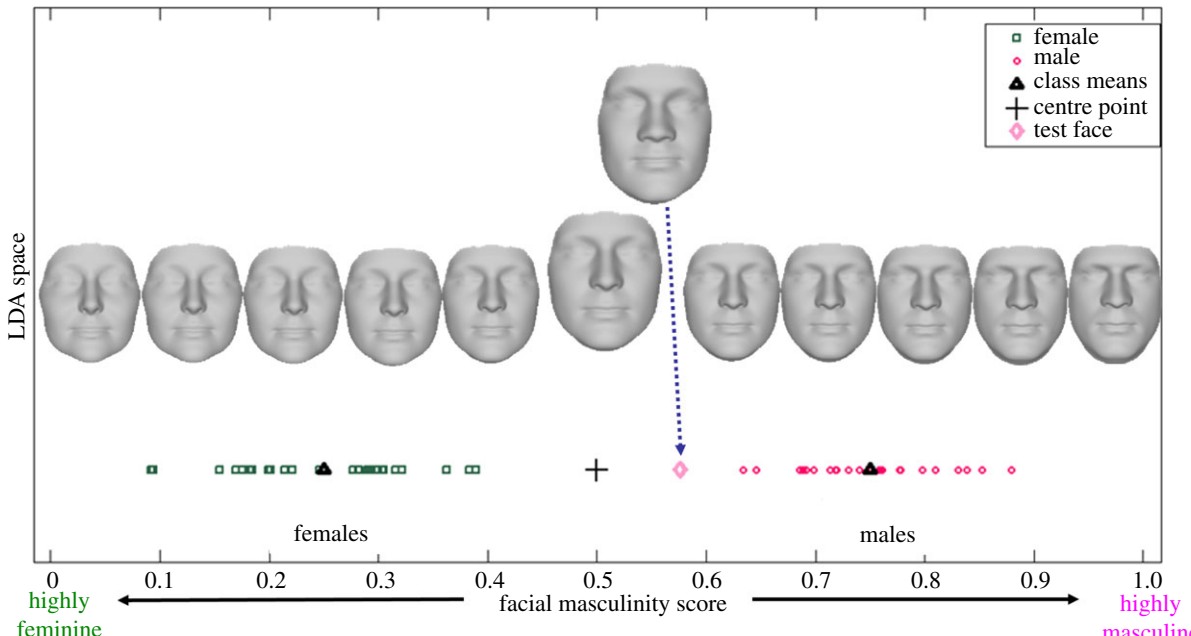

**Figure 2.** Calculation of a facial masculinity score for each face. The 10 features selected by the GEFS algorithm were projected in the LDA space, which divides the sample into two classes of males and females. The mean of each class (marked by the black triangles) was used to identify the mean of the two classes (marked by the black cross). The masculinity score was calculated for each face by projecting its feature vector on the LDA space and using the distance between this projection and the mean of the two classes. These distances were then scaled to give values between 0 (highly feminine) and 1 (highly masculine). The composite images shown in the figure reflect changes in facial structure modelled according to the varying degrees of facial masculinity, using the current set of images. For a more detailed description of this general approach, see [21]. (Online version in colour.)

## (d) Statistical analyses

All statistical analyses were conducted using R and RStudio [33,34]. All continuous variables were found to be normally distributed except for two facial distances (geodesic forehead width and geodesic mandible height) which were positively skewed and hence were log-transformed for analyses. Potential confounding effects of age and facial area on the facial masculinity scores and distances were examined using Pearson's correlation analyses. A 2 (family group: parents of autistic versus non-autistic children) by 2 (sex: males versus females) analysis of variance was conducted on the facial masculinity scores and distances. If age or facial area correlated significantly with facial masculinity or distance scores, these variables were included in a subsequent analysis of covariance (ANCOVA) model. Bonferroni correction was applied to account for multiple testing; thus, an alpha level of 0.005 was used to determine statistical significance. Additionally, we conducted a further LDA to examine the accuracy of the facial variables derived from parents' facial images in classifying the autism diagnostic status of their children.

## 3. Results

### (a) Feature selection and gender classification accuracy

The GEFS algorithm selected four Euclidean (nasal width, nasal tip protrusion, nasal bridge length and upper lip height) and six geodesic (upper facial height, outer canthal width, forehead width, mandible height, and left and right upper cheek heights) distances as the most discriminating facial features between males and females. Based on the 10 selected features, the LDA classifier was found to correctly classify male and female facial images with 95.7% accuracy.

### (b) Evaluating potential confounding variables

Pearson correlation analyses revealed that facial area (parents of autistic children: $M = 35\,435$ mm$^2$, s.d. $= 4320$; parents of

non-autistic children: $M = 35\,142$ mm$^2$, s.d. $= 4248$) and age of participants significantly correlated with facial masculinity scores and distances ($0.12 \le r \le 0.56$, all $p < 0.05$, see electronic supplementary material, table S1 for full correlation matrix). Therefore, we controlled for effects of facial area and age in all subsequent analyses.

### (c) Group comparisons

Table 2 presents descriptive statistics for facial masculinity scores and facial distances of male and female parents of autistic or non-autistic children. The main effects of family group (parents of autistic versus non-autistic children) and sex (males versus females) on the facial masculinity scores and distances were examined in several ANCOVA models (see table 2 for the test statistics for the main effect of family group). Briefly, the analyses showed that parents of autistic children presented with greater facial masculinity than parents of non-autistic children, both in terms of their facial masculinity scores (small effect) and facial distances (all distances with small-to-medium effect, except for four distances: nasal width, upper lip height, mandible height and left upper cheek height). Figure 3 presents probability density functions for the facial masculinity scores across the four groups of participants. The graph shows a shift towards increased facial masculinization among fathers and mothers of children on the autism spectrum.

As expected, there were large effects of sex where males presented with significantly more masculinized facial structures than females, both in terms of their facial masculinity scores ($d = 4.69$; very large effect) and facial distances ($0.74 \le d \le 1.38$; large effect; see electronic supplementary material, table S2 for descriptive and test statistics). The group × sex interaction effects were not statistically significant for all analyses (all $p > 0.10$), indicating that both fathers and

Table 2. Descriptive and null hypothesis testing statistics for the effect of family group (parents of autistic versus non-autistic children) on facial masculinity scores and facial distances (in mm).

| variables | parents of autistic children (n = 192) | | | | parents of non-autistic children (n = 163) | | | | test statistics[a] | | | |
| | fathers (n = 58) | | mothers (n = 134) | | males (n = 50) | | females (n = 113) | | | | | |
| | M | s.d. | M | s.d. | M | s.d. | M | s.d. | F(1,349) | p | d | 95% CI |
|---|---|---|---|---|---|---|---|---|---|---|---|---|
| masculinity score | 0.85 | 0.08 | 0.32 | 0.12 | 0.80 | 0.11 | 0.28 | 0.09 | 17.3 | <0.001 | 0.16 | [0.01, 0.35] |
| Euclidean distances | | | | | | | | | | | | |
| nasal width | 19.8 | 1.86 | 18.2 | 1.84 | 19.3 | 2.11 | 17.6 | 1.88 | 7.31 | 0.007 | 0.27 | [0.06, 0.48] |
| nasal tip protrusion | 21.1 | 2.05 | 17.2 | 2.22 | 19.5 | 2.62 | 16.2 | 2.50 | 23.7 | <0.001 | 0.41 | [0.20, 0.63] |
| nasal bridge length | 51.0 | 3.86 | 46.3 | 3.66 | 49.3 | 2.83 | 45.4 | 3.29 | 9.34 | 0.002 | 0.28 | [0.06, 0.48] |
| upper lip height | 25.5 | 3.11 | 23.1 | 2.73 | 24.8 | 3.04 | 22.7 | 2.71 | 2.94 | 0.08 | 0.17 | [0.01, 0.38] |
| geodesic distances | | | | | | | | | | | | |
| upper facial height | 73.0 | 4.57 | 67.7 | 4.57 | 70.7 | 4.51 | 65.5 | 4.58 | 15.5 | <0.001 | 0.35 | [0.14, 0.56] |
| outer canthal width | 120.6 | 9.03 | 109.8 | 7.82 | 117.3 | 8.96 | 106.9 | 7.92 | 11.7 | <0.001 | 0.31 | [0.11, 0.52] |
| forehead width | 167.3 | 11.5 | 153.1 | 8.04 | 160.7 | 11.7 | 150.1 | 7.20 | 19.6 | <0.001 | 0.38 | [0.17, 0.59] |
| mandible height | 77.7 | 5.50 | 71.8 | 4.79 | 76.0 | 7.67 | 71.7 | 4.77 | 1.43 | 0.23 | 0.11 | [0, 0.31] |
| upper cheek height (left) | 80.0 | 5.06 | 73.9 | 3.80 | 77.9 | 4.00 | 73.2 | 3.95 | 7.75 | 0.006 | 0.01 | [0.01, 0.44] |
| upper cheek height (right) | 79.6 | 4.40 | 73.9 | 3.59 | 77.4 | 4.56 | 72.7 | 3.81 | 14.7 | <0.001 | 0.27 | [0.06, 0.48] |

[a]ANCOVA model adjusted for facial area and age. Alpha levels adjusted for multiple testing ($\alpha = 0.005$).

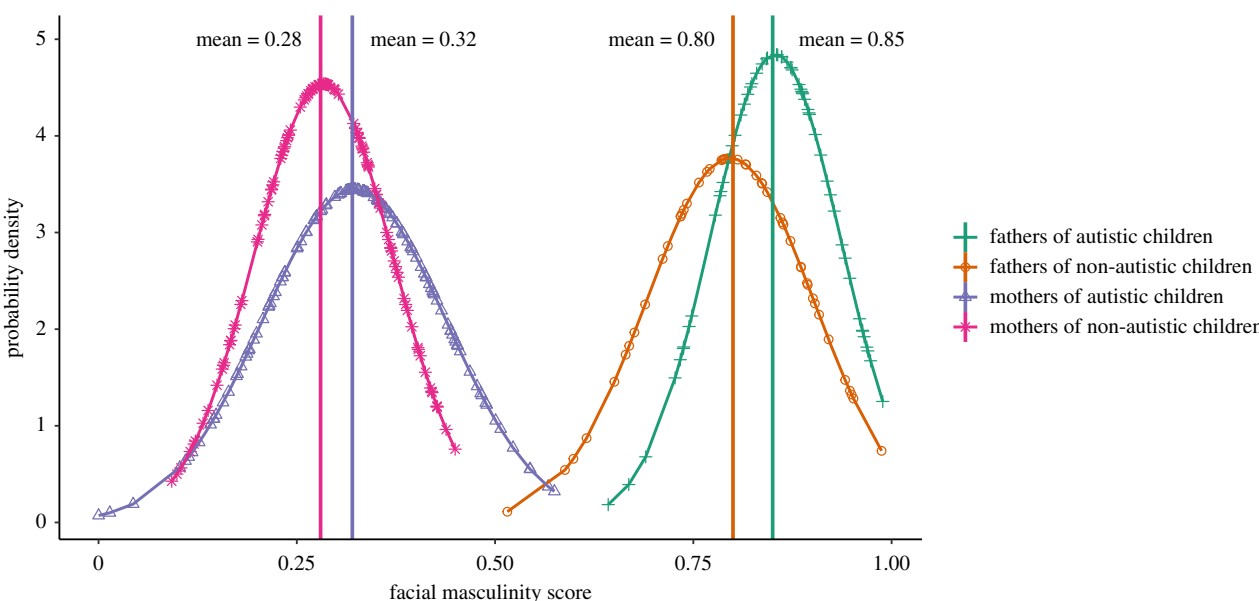

**Figure 3.** Probability density functions that show the distributions of facial masculinity scores across the four groups of participants. (Online version in colour.)

mothers of autistic children showed more masculinized facial structures than their same-sex counterparts.

## (d) Classification of autism diagnostic status

We conducted an LDA with 10-fold cross-validation to examine the accuracy of the seven facial variables found to be statistically significantly different between the two groups of parents in correctly classifying the autism diagnosis status of their children. The seven facial variables entered into the LDA model were facial masculinity scores, nasal tip protrusion, upper lip height, upper facial height, outer canthal width, forehead width and right upper cheek height. Based on these features, the LDA classifier was found to correctly classify the presence of an autism diagnosis with 74.3% accuracy.

## 4. Discussion

We investigated whether a broad autism phenotype expressed as increased facial masculinity is found among parents of autistic children relative to age-matched parents without a family history of autism. Consistent with our hypothesis, we found that facial masculinity was generally more pronounced in both fathers and mothers of autistic children compared to their same-sex counterparts. These findings complement previous reports that facial features of autistic children were more masculinized than the features of non-autistic children [22] and that non-autistic siblings of autistic children also presented with greater facial masculinity than non-autistic children without autistic siblings [7]. Taken together, this research provides evidence that facial masculinity may be a heritable broad autism phenotype. Furthermore, the evidence of facial masculinization for both fathers and mothers of autistic children supports a genetic aetiology for this physical feature, importantly, with both paternal and maternal lineages implicated.

In the current study, six facial distances were found to distinguish between parents of autistic children and parents of non-autistic children. Of the six features, two—increased outer canthal width and forehead width—were previously observed

to distinguish autistic children [22] and non-autistic siblings of autistic children [7] from their respective comparison groups. The four remaining features—increased nasal tip protrusion, nasal bridge length, upper facial height and right upper cheek height—appear to be unique to differentiating the two parent groups. These differences in distinguishing facial features across studies probably reflect the substantial puberty-related changes in the way sexual dimorphism in facial structure is expressed in post-pubertal adults compared to prepubescent children [35]. Four of the six features that distinguished the two parent groups were selected by GEFS in an earlier study to distinguish the faces of male and female young adults [19].

The increased facial masculinity among parents of autistic children observed in the current study is consistent with the key outcomes of the previous studies involving autistic children [22] and their siblings [7]. In all of these studies, differences in facial masculinity as a function of autism or familial relationship were present for both males and females. Intriguingly, however, these results run counter to findings reported in two of our previous studies where we investigated facial masculinity among non-autistic adults with high or low levels of autistic traits [36,37]. In both studies, we found that females with high levels of autistic traits presented more masculinized facial structures than females with low trait levels, whereas males with high trait levels showed *less* masculinized facial features than males with low trait levels. This set of observations is consistent with an earlier suggestion that the morphology of autistic adults is better described as being androgynous rather than hypermasculinized [38]. Since non-autistic parents of autistic children typically report high levels of autistic traits (e.g. [39]), it is unclear why parents included in the current study showed hypermasculinized rather than androgynous facial structures. One possibility is that androgynous facial features associated with sub-clinical autistic traits in the general population may be linked to biological mechanisms that are somewhat different to the genetic markers of autism. This assertion could be tested in further research examining facial masculinity of adults with a clinical diagnosis of autism.

We conducted further analysis to examine the accuracy of the seven critical facial variables in correctly classifying the

autism diagnosis status of the parents' children. Our analysis showed that classification accuracy was 74.3%. One reason for the modest classification performance is that facial masculinity may represent only one of several facial phenotypes associated with autism. In a parallel line of research, our group found greater facial asymmetry among autistic children [40] as well as in parents of autistic children [6], compared to matched groups without a family history of autism. Thus, classification accuracy potentially could be improved by adding other predictors such as facial asymmetry into the LDA model. Including predictors derived from the facial images of siblings may also improve classification accuracy.

The key strengths of the present study include the large sample size of parents of autistic children and comparison adults recruited from the general population, and the use of three-dimensional facial photogrammetry and sophisticated facial analytic techniques for examining facial masculinity. Nevertheless, there are two limitations to be borne in mind when interpreting the current findings. First, we did not have access to separate datasets for training and testing our classification algorithm, and thus have relied on iterative division of our samples into independent training and testing sets. While large open-source data are available, these datasets do not typically include information regarding family history of autism. Thus, future replication studies are warranted. Second, our participants were restricted to those of Caucasian descent. Therefore, it is unclear whether the current observations will extend to people of ethnically diverse backgrounds. As autism is diagnosed at similar rates regardless of ethnicity [41], it will be imperative to build on the current programme of research to investigate whether facial masculinization associated with autism generalizes to other ethnic populations.

## 5. Conclusion

The current findings contribute substantially to our understanding of broad autism phenotype in two ways. First, the current results add to an accumulating evidence base that the broad autism phenotype observed among first-degree relatives of autistic people is expressed not only in behaviour and cognition but also in facial characteristics. Second, given

that both fathers and mothers of autistic children presented increased facial masculinity, this suggests that facial masculinization previously observed among children on the autism spectrum and their siblings indicates potential heritability of this phenotype from both parents. This finding is consistent with large population-based data suggesting that the occurrence of autism is likely to be connected to both paternal and maternal lineages [42]. A potential avenue for future research is to compare facial masculinity between members of simplex (one immediate family member diagnosed with autism) and multiplex (more than one family members diagnosed with autism) families to examine whether effects of masculinity increase as a function of genetic liability for autism.

Ethics. All participants provided written informed consent, and ethics approval was granted by the Human Research Ethics Committee of the University of Western Australia (RA/4/1/5657).

Data accessibility. Data and analysis codes are available at https://figshare.com/articles/dataset/parents-face-masc/19217013 and https://github.com/dianawtan/parents-face-masc.

Authors' contributions. D.W.T.: conceptualization, formal analysis, visualization, writing—original draft, writing—review and editing; S.Z.G.: formal analysis, methodology, software, visualization, writing—review and editing; G.A.A.: data curation, writing—review and editing; A.M.: software, supervision, writing—review and editing; A.J.O.W.: funding acquisition, supervision, writing—review and editing; M.T.M.: conceptualization, formal analysis, funding acquisition, supervision, writing—review and editing.

All authors gave final approval for publication and agreed to be held accountable for the work performed therein.

Competing interests. We declare we have no competing interests.

Funding. The authors acknowledge funding support from Telethon Kids Institute for the collection of data from adults without a family history of ASD. D.W.T. was supported by research funding provided by the Faculty of Science, University of Western Australia allocated to M.T.M. A.J.O.W. is supported by a National Health and Medical Research Council Senior Research Fellowship (APP 1077966).

## Endnotes

[1]Many people on the autism spectrum prefer 'identity-first language' such as 'autistic children' or 'children on the autism spectrum' [3]. Therefore, we adopted identity-first language throughout this article, where appropriate.

[2]This sample was included in a study of facial asymmetry reported by Tan et al. [6].

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
