## [Peer Review File · Proceedings of the Royal Society B: Biological Sciences]

Review History

RSPB-2021-1381.R0 (Original submission)

Review form: Reviewer 1

Recommendation

Major revision is needed (please make suggestions in comments)

Scientific importance: Is the manuscript an original and important contribution to its field?

Marginal

General interest: Is the paper of sufficient general interest?

Acceptable

Quality of the paper: Is the overall quality of the paper suitable?

Acceptable

Is the length of the paper justified?

No

Should the paper be seen by a specialist statistical reviewer?

No

Do you have any concerns about statistical analyses in this paper? If so, please specify them explicitly in your report.

No

It is a condition of publication that authors make their supporting data, code and materials available - either as supplementary material or hosted in an external repository. Please rate, if applicable, the supporting data on the following criteria.

Is it accessible?

Yes

Is it clear?

Yes

Is it adequate?

Yes

Do you have any ethical concerns with this paper?

No

Comments to the Author

Tan and colleagues present a continuation of their previous work on a broad autism phenotype expressed as masculinity of the face (e.g., Tan, et al. 2020). Previously this team has shown, among other results, that siblings of autistic individuals tend to have more masculine faces. The current (very brief) study finds that parents of autistic children also tend to have more masculine faces, regardless of the sex of the parent. The finding is believable and congruent with both the past work of this team and research by others. However, the reader is left with an impression, difficult to shake, that the authors are slicing infinitesimally small pieces off of their existing data set on this topic and publishing one finding at a time. The paper, its figures, and its data are highly similar to previous work, with the sole difference being that this paper is about the parents. It's unclear to me that this finding on its own warrants an entirely distinct scholarly work. The topic itself is important, but the finding that the parents also partake in this broader autism phenotype is largely unsurprising, and on its own does not meaningfully refine or deepen our understanding. Consequently, the major critique is simply that the study needs more meat.

Other comments:

* what does a "masculine" face look like in comparison to an androgynous, feminine, or other face? The current figures provide little insight into what features the algorithm relies on (and how) in making this determination

* the authors invoke genetics of facial masculinity in the discussion (and is an implicit theme of the premise), but there was not much context given or connections of this hypothesis to the broader literature on this topic, e.g., PMID: 24213680, PMID: 33288918, PMID: 34108004, and others

* was the LDA classifier trained only on this adult cohort?

* what was the sex breakdown of the autistic children? Do the parents of autistic males have more masculine faces than the parents of autistic females?

* Figure 2 could be improved by having lines (and the value) of the group means

Review form: Reviewer 2

Recommendation

Major revision is needed (please make suggestions in comments)

Scientific importance: Is the manuscript an original and important contribution to its field?

Good

General interest: Is the paper of sufficient general interest?

Good

Quality of the paper: Is the overall quality of the paper suitable?

Marginal

Is the length of the paper justified?

Yes

Should the paper be seen by a specialist statistical reviewer?

No

Do you have any concerns about statistical analyses in this paper? If so, please specify them explicitly in your report.

Yes

It is a condition of publication that authors make their supporting data, code and materials available - either as supplementary material or hosted in an external repository. Please rate, if applicable, the supporting data on the following criteria.

Is it accessible?

Yes

Is it clear?

Yes

Is it adequate?

No

Do you have any ethical concerns with this paper?

No

Comments to the Author

The findings raise interesting questions about possible links between ASD and factors influencing masculinity. The measure of facial masculinity is central to the report, but is described at a very abstract level, eg in terms of 20 unspecified "distances", p5, and I'm not clear on why this approach was chosen. I would ask for further information and testing before acceptance, as described below. Many of my concerns arise because there is little or maybe no investigation of the underlying masculinity differences driving the effects.

First, I am curious why the authors do not use a more conventional GMM approach to shape analysis, an initial procrustean fit followed by a sex classifier on a training set considering the coordinates of all landmarks, and then using the trained classifier to generate a predicted sex-classification for the experimental set of parents and controls. This would seem to have important advantages: (1) it would consider the position and relative distances of all landmarks within the shape (not eg the 20 selected by the experimenters), and (2) it would also produce a size measure for each shape. A possible (3) is that this approach is widely used. Finally (4) this might allow

visualisation of the group (parent vs control) face differences.

Second, it is of course best practice to train the classifier (in whatever form) on a different sample than that used to test the classifier results. In this case (3d scans of men and women) there isn't a shortage of materials available, and this would be preferable to the 10-fold validation currently used.

Third, the automated landmark process used here does not seem to capture any of the facial outline, particularly the jaw. This is strange as this is the jaw and brow are key regions for human dimorphism. Shouldn't the jawline be included?

Fourth, men and women seem to differ in their mouth shape at "neutral", where men present slightly less evidence of a smile than women. This might possibly drive the masculinity score, yet we don't know from the present analyses. Whether a masculinity difference is driven by a potentially controllable cue, such as expression, or by a more biologically driven one, such as brow distance, would be very interesting to know. So again I highly encourage a more thorough investigation of the shape differences driving any effect.

Fifth, the confound of "facial area" and masculinity is unfortunate. I don't believe it is possible to tell from the report whether the classification algorithm is simply using some aspect of distance. For example, if parents of autistic children weighed more than the controls, and therefore had wider faces, would this be sufficient to generate a "masculinity" difference using the reported analyses?

Therefore I would suggest a more conventional GMM approach to generating a predicted masculinity score, based on a sample outside the experimental group, and which then looks backward to see where and what the masculinity differences are that drive any parent-control difference.

Review form: Reviewer 3

Recommendation

Accept with minor revision (please list in comments)

Scientific importance: Is the manuscript an original and important contribution to its field?

Good

General interest: Is the paper of sufficient general interest?

Good

Quality of the paper: Is the overall quality of the paper suitable?

Good

Is the length of the paper justified?

Yes

Should the paper be seen by a specialist statistical reviewer?

No

Do you have any concerns about statistical analyses in this paper? If so, please specify them explicitly in your report.

No

It is a condition of publication that authors make their supporting data, code and materials available - either as supplementary material or hosted in an external repository. Please rate, if applicable, the supporting data on the following criteria.

Is it accessible?

N/A

Is it clear?

N/A

Is it adequate?

N/A

Do you have any ethical concerns with this paper?

No

Comments to the Author

This is a very interesting and important study showing masculinised facial features in parents of autistic children. The findings are fascinating and the methods are appropriate.

In the abstract, it would be clearer if the authors could use Cohen's d instead of partial eta squared because the factors and covariates included in the ANCOVA model are not listed in the abstract.

In the introduction, it would be useful if the authors could go into a bit more detail regarding the findings from the Danish biobank cohort. There was no relationship between amniotic testosterone and autism, and only males were included in the study. It would also be useful to include more recent literature showing no relationship between early androgen exposure and autistic traits (e.g., Kung et al., in press, JCPP).

As for participants, were those control adults parents as well? How was family ASD history assessed?

Regarding analyses, is it possible to include a set of analyses focusing on specific facial landmarks/features in supplementary materials?

In the discussion, the inconsistencies regarding the links amongst early androgen exposure, autism/autistic traits, and facial masculinity need to be acknowledged. Early androgen exposure seems to relate to facial masculinity but not autism/autistic traits. I do appreciate that it is very challenging to explain and integrate these complex findings in the literature. However, at the very least, I think the inconsistencies should be acknowledged. Also, has there been any research comparing facial structures/features between autistic and neurotypical adults? Finally, are there any other confounding variables, other than age and facial area, that could have contributed to the observed group difference?

Congratulations to the authors for completing this piece of significant research!

Decision letter (RSPB-2021-1381.R0)

11-Aug-2021

Dear Dr Tan:

I have now received three reviews of your manuscript and discussed it with the Associate Editor. I am writing to inform you that based on these assessments and my own review, your manuscript RSPB-2021-1381 entitled "An investigation of a novel broad autism phenotype: Increased facial masculinity among parents of children on the autism spectrum" has, in its current form, been rejected for publication in Proceedings B. However, despite the reviewers' reservations, the AE and I also agree with them that your premise is very interesting. Therefore, we would be happy to consider a resubmission, provided the comments of the referees are fully addressed. Most importantly, we find the most compelling part of your study the potential for prediction, however this is not sufficiently developed. We would like for you to evaluate your ability to predict autistic children on the features of the parents (for instance, using the algorithms/approaches suggested by reviewer 2). This may also help you address Reviewer 1's concern that this study does not represent a notable advance on previous work. In addition, Reviewer 2 suggests an alternate analysis, which we would like to see, or justification for your current approach that takes into account the reviewer's concerns (such as whether the controls were parents, any weight difference between the groups, what features were included, etc). Finally, your manuscript as written doesn't meet Proceedings B's mandate for having broad biological relevance, although I think you could address this by, for instance, considering why this phenotype may have been selected, and why it is associated with increased masculinization, among other possibilities. Please note that this is not a provisional acceptance; we will solicit reviews for your revision before making a new decision on the manuscript.

Sincerely,
Dr Sarah Brosnan
Editor, Proceedings B
mailto: proceedingsb@royalsociety.org

Associate Editor
Board Member: 1
Comments to Author:

Following the comments from two reviewers, shortcomings were noted in the manuscript. While not necessarily fatal flaws, we think that the comments may take more than a few months to address. Therefore, we have assessed the manuscript as 'Reject and allow resubmission'.

One reviewer noted that the manuscript should have more depth. In this editor's opinion, this could be achieved by attempting to develop a metric by which the facial measures of the parents anticipate the autistic status of the children. Essentially, something that could theoretically be brought into any clinic or doctor's office.

We hope that you are able to deal with the methodological concerns raised by the reviewers and to see your manuscript in the future.

Reviewer(s)' Comments to Author:

Referee: 1

Comments to the Author(s)

Tan and colleagues present a continuation of their previous work on a broad autism phenotype expressed as masculinity of the face (e.g., Tan, et al. 2020). Previously this team has shown, among other results, that siblings of autistic individuals tend to have more masculine faces. The current (very brief) study finds that parents of autistic children also tend to have more masculine faces, regardless of the sex of the parent. The finding is believable and congruent with both the past work of this team and research by others. However, the reader is left with an impression, difficult to shake, that the authors are slicing infinitesimally small pieces off of their existing data set on this topic and publishing one finding at a time. The paper, its figures, and its data are highly similar to previous work, with the sole difference being that this paper is about the parents. It's unclear to me that this finding on its own warrants an entirely distinct scholarly work. The topic itself is important, but the finding that the parents also partake in this broader autism phenotype is largely unsurprising, and on its own does not meaningfully refine or deepen our understanding. Consequently, the major critique is simply that the study needs more meat.

Other comments:

- * what does a "masculine" face look like in comparison to an androgynous, feminine, or other face? The current figures provide little insight into what features the algorithm relies on (and how) in making this determination
- * the authors invoke genetics of facial masculinity in the discussion (and is an implicit theme of the premise), but there was not much context given or connections of this hypothesis to the broader literature on this topic, e.g., PMID: 24213680, PMID: 33288918, PMID: 34108004, and others
- * was the LDA classifier trained only on this adult cohort?
- * what was the sex breakdown of the autistic children? Do the parents of autistic males have more masculine faces than the parents of autistic females?
- * Figure 2 could be improved by having lines (and the value) of the group means

Referee: 2

Comments to the Author(s)

The findings raise interesting questions about possible links between ASD and factors influencing masculinity. The measure of facial masculinity is central to the report, but is described at a very abstract level, eg in terms of 20 unspecified "distances", p5, and I'm not clear on why this approach was chosen. I would ask for further information and testing before acceptance, as described below. Many of my concerns arise because there is little or maybe no investigation of the underlying masculinity differences driving the effects.

First, I am curious why the authors do not use a more conventional GMM approach to shape analysis, an initial procrustean fit followed by a sex classifier on a training set considering the coordinates of all landmarks, and then using the trained classifier to generate a predicted sex-classification for the experimental set of parents and controls. This would seem to have important advantages: (1) it would consider the position and relative distances of all landmarks within the shape (not eg the 20 selected by the experimenters), and (2) it would also produce a size measure

for each shape. A possible (3) is that this approach is widely used. Finally (4) this might allow visualisation of the group (parent vs control) face differences.

Second, it is of course best practice to train the classifier (in whatever form) on a different sample than that used to test the classifier results. In this case (3d scans of men and women) there isn't a shortage of materials available, and this would be preferable to the 10-fold validation currently used.

Third, the automated landmark process used here does not seem to capture any of the facial outline, particularly the jaw. This is strange as this is the jaw and brow are key regions for human dimorphism. Shouldn't the jawline be included?

Fourth, men and women seem to differ in their mouth shape at "neutral", where men present slightly less evidence of a smile than women. This might possibly drive the masculinity score, yet we don't know from the present analyses. Whether a masculinity difference is driven by a potentially controllable cue, such as expression, or by a more biologically driven one, such as brow distance, would be very interesting to know. So again I highly encourage a more thorough investigation of the shape differences driving any effect.

Fifth, the confound of "facial area" and masculinity is unfortunate. I don't believe it is possible to tell from the report whether the classification algorithm is simply using some aspect of distance. For example, if parents of autistic children weighed more than the controls, and therefore had wider faces, would this be sufficient to generate a "masculinity" difference using the reported analyses?

Therefore I would suggest a more conventional GMM approach to generating a predicted masculinity score, based on a sample outside the experimental group, and which then looks backward to see where and what the masculinity differences are that drive any parent-control difference.

Referee: 3

Comments to the Author(s)

This is a very interesting and important study showing masculinised facial features in parents of autistic children. The findings are fascinating and the methods are appropriate.

In the abstract, it would be clearer if the authors could use Cohen's d instead of partial eta squared because the factors and covariates included in the ANCOVA model are not listed in the abstract.

In the introduction, it would be useful if the authors could go into a bit more detail regarding the findings from the Danish biobank cohort. There was no relationship between amniotic testosterone and autism, and only males were included in the study. It would also be useful to include more recent literature showing no relationship between early androgen exposure and autistic traits (e.g., Kung et al., in press, JCPP).

As for participants, were those control adults parents as well? How was family ASD history assessed?

Regarding analyses, is it possible to include a set of analyses focusing on specific facial landmarks/features in supplementary materials?

In the discussion, the inconsistencies regarding the links amongst early androgen exposure, autism/autistic traits, and facial masculinity need to be acknowledged. Early androgen exposure seems to relate to facial masculinity but not autism/autistic traits. I do appreciate that it is very challenging to explain and integrate these complex findings in the literature. However, at the

very least, I think the inconsistencies should be acknowledged. Also, has there been any research comparing facial structures/features between autistic and neurotypical adults? Finally, are there any other confounding variables, other than age and facial area, that could have contributed to the observed group difference?

Congratulations to the authors for completing this piece of significant research!

Author's Response to Decision Letter for (RSPB-2021-1381.R0)

See Appendix A.

RSPB-2022-0143.R0

Review form: Reviewer 1

Recommendation

Major revision is needed (please make suggestions in comments)

Scientific importance: Is the manuscript an original and important contribution to its field?

Acceptable

General interest: Is the paper of sufficient general interest?

Good

Quality of the paper: Is the overall quality of the paper suitable?

Good

Is the length of the paper justified?

Yes

Should the paper be seen by a specialist statistical reviewer?

No

Do you have any concerns about statistical analyses in this paper? If so, please specify them explicitly in your report.

No

It is a condition of publication that authors make their supporting data, code and materials available - either as supplementary material or hosted in an external repository. Please rate, if applicable, the supporting data on the following criteria.

Is it accessible?

Yes

Is it clear?

Yes

Is it adequate?

Yes

Do you have any ethical concerns with this paper?

No

Comments to the Author

The substantially revised introduction, the reworked figures, and the additional analyses performed by the authors were all responsive to my original critiques, and I have no further critiques. The current manuscript now does a better job of justifying itself and the associated work in the overall trajectory of this line of research. Well done.

Decision letter (RSPB-2022-0143.R0)

18-Feb-2022

Dear Dr Tan

I am pleased to inform you that your Review manuscript RSPB-2022-0143 entitled "An investigation of a novel broad autism phenotype: Increased facial masculinity among parents of children on the autism spectrum" has been accepted for publication in Proceedings B.

The referee(s) do not recommend any further changes. Therefore, please proof-read your manuscript carefully and upload your final files for publication. Because the schedule for publication is very tight, it is a condition of publication that you submit the revised version of your manuscript within 7 days. If you do not think you will be able to meet this date please let me know immediately.

To upload your manuscript, log into <http://mc.manuscriptcentral.com/prsb> and enter your Author Centre, where you will find your manuscript title listed under "Manuscripts with Decisions." Under "Actions," click on "Create a Revision." Your manuscript number has been appended to denote a revision.

You will be unable to make your revisions on the originally submitted version of the manuscript. Instead, upload a new version through your Author Centre.

- 1) A text file of the manuscript (doc, txt, rtf or tex), including the references, tables (including captions) and figure captions. Please remove any tracked changes from the text before submission. PDF files are not an accepted format for the "Main Document".
- 2) A separate electronic file of each figure (tiff, EPS or print-quality PDF preferred). The format should be produced directly from original creation package, or original software format. Please note that PowerPoint files are not accepted.
- 3) Electronic supplementary material: this should be contained in a separate file from the main text and the file name should contain the author's name and journal name, e.g `authorname_procb_ESM_figures.pdf`

All supplementary materials accompanying an accepted article will be treated as in their final form. They will be published alongside the paper on the journal website and posted on the online figshare repository. Files on figshare will be made available approximately one week before the accompanying article so that the supplementary material can be attributed a unique DOI. Please see: <https://royalsociety.org/journals/authors/author-guidelines/>

- 4) Data-Sharing and data citation

It is a condition of publication that data supporting your paper are made available. Data should be made available either in the electronic supplementary material or through an appropriate repository. Details of how to access data should be included in your paper. Please see <https://royalsociety.org/journals/ethics-policies/data-sharing-mining/> for more details.

If you wish to submit your data to Dryad (<http://datadryad.org/>) and have not already done so you can submit your data via this link <http://datadryad.org/submit?journalID=RSPB&manu=RSPB-2022-0143> which will take you to your unique entry in the Dryad repository.

Once again, thank you for submitting your manuscript to Proceedings B and I look forward to receiving your final version. If you have any questions at all, please do not hesitate to get in touch.

Sincerely,
Dr Sarah Brosnan
Editor, Proceedings B
<mailto:proceedingsb@royalsociety.org>

Associate Editor
Comments to Author:

Thank you for your submission. We are pleased to inform you that your article is accepted with minor revisions. Please look for further communication.

Reviewer(s)' Comments to Author:

Referee: 1

Comments to the Author(s).

The substantially revised introduction, the reworked figures, and the additional analyses performed by the authors were all responsive to my original critiques, and I have no further critiques. The current manuscript now does a better job of justifying itself and the associated work in the overall trajectory of this line of research. Well done.

Decision letter (RSPB-2022-0143.R1)

23-Feb-2022

Dear Dr Tan

I am pleased to inform you that your manuscript entitled "An investigation of a novel broad autism phenotype: Increased facial masculinity among parents of children on the autism spectrum" has been accepted for publication in Proceedings B.

Your article has been estimated as being 8 pages long. Our Production Office will be able to confirm the exact length at proof stage.

Data Accessibility section

Open Access

Paper charges

Sincerely,

Proceedings B

Dear Reviewers,

Thank you for your insightful and constructive feedback on our manuscript titled *An investigation of a novel broad autism phenotype: Increased facial masculinity among parents of children on the autism spectrum*. Please find our point-by-point response to each comment below. Where appropriate, changes have been made based on your recommendations which resulted in a manuscript that now more clearly outlines the broad biological relevance of our work, as well as providing more detailed reporting of our analytical approach.

Reviewer 1

1. Tan and colleagues present a continuation of their previous work on a broad autism phenotype expressed as masculinity of the face (e.g., Tan, et al. 2020). Previously this team has shown, among other results, that siblings of autistic individuals tend to have more masculine faces. The current (very brief) study finds that parents of autistic children also tend to have more masculine faces, regardless of the sex of the parent. The finding is believable and congruent with both the past work of this team and research by others. However, the reader is left with an impression, difficult to shake, that the authors are slicing infinitesimally small pieces off of their existing data set on this topic and publishing one finding at a time. The paper, its figures, and its data are highly similar to previous work, with the sole difference being that this paper is about the parents. It's unclear to me that this finding on its own warrants an entirely distinct scholarly work. The topic itself is important, but the finding that the parents also partake in this broader autism phenotype is largely unsurprising, and on its own does not meaningfully refine or deepen our understanding. Consequently, the major critique is simply that the study needs more meat.

Response:

In this comment, Reviewer 1 has raised several points which we will respond to in turn. First, the Reviewer described how our current findings were analysed and presented in a way that is highly similar to our previous studies (Tan et al., 2017 and Tan et al., 2020a). Our research team made a considered decision to maintain consistency in data analyses and reporting to allow us to draw direct comparisons across the different datasets. There are several reasons why these data were not all published in one paper. First, these datasets (i.e., the facial images of autistic children, siblings, and parents) were not available at the same time. Facial images for the three groups were collected in combination with collecting data for various other unrelated projects. Second, the first two papers and the current manuscript build substantially on each other. The Tan et al. (2017) paper was the first to report a masculinised facial phenotype for autistic children. Next, in Tan et al. (2020a), we reported that siblings of autistic children also demonstrated a masculinised facial structure, consistent with a unique physical manifestation of the broad autism phenotype. Note that in Tan et al. (2020a) we also had access to additional facial images of non-autistic children without a family history of autism which allowed us to test the generalisability of the gender

classification algorithm developed as part of the Tan et al. (2017) paper. The present manuscript is a significant extension of the previous papers. First, it provides important confirmation that a masculinised facial structure characterises families with autism, with this characteristic extending to parents. We are not aware of this level of consistency in evidence for any other physical characteristic associated with autism. Second, because both the mothers and fathers of autistic children showed a masculinised facial structure, this provides critical evidence that there is likely to be both maternal and paternal lines of genetic transmission for this feature.

We agree that parents' broad autism phenotype expressed as facial masculinisation is theoretically expected given the heritability of both autistic and facial characteristics. This is, however, the first study that provides empirical data to test this assertion. Thus, we think it is important to publish these data, particularly in the field of autism research where the field has traditionally focussed on publishing 'novel' findings that do not tend to be replicated (see Muller and Amaral's 2017 Editorial, [doi: 10.1002/aur.1746](https://doi.org/10.1002/aur.1746)).

However, we do agree with the reviewer that the initial manuscript lacked substance. Accordingly, in the revised manuscript we now provide a more extensive theoretical rationale that better articulates the value in studying facial masculinity in the parents of autistic children. The abstract and discussion have also been bolstered in this respect.

Further, we also made two other revisions:

- a) We included several analyses of facial features that characterise facial masculinity in this sample (see lines 241-254 and Table 2 on pg. 15).
- b) On the Editor's advice, we added a predictive analysis to examine how accurately parents' facial masculinity could predict the autism status of their children (see lines 267-274).

2. What does a "masculine" face look like in comparison to an androgynous, feminine, or other face? The current figures provide little insight into what features the algorithm relies on (and how) in making this determination.

Response:

3. Note that in preparing to respond to this comment, we realised there was a mistake in the initial manuscript where we referred to the measurement of 21 Euclidean and 20 geodesic distances. Instead, 26 Euclidean and 26 geodesic distances were used. Returning to this comment, the 26 Euclidean and 26 geodesic distances were measured between 21 landmarks and for clarity we now list these distances in the newly added Table 1 (on pg. 10). These distances were entered into the Gradient-based Efficient Feature Selection (GEFS) algorithm which selected a set of 10 distances that maximally contributed to the overall accuracy in classifying the facial images according to sex. The 10 distances were four Euclidean distances (nasal width, nasal tip protrusion, nasal bridge length, and upper lip length) and six geodesic distances (upper facial height, outer canthal width, forehead width,

mandible height, and left and right upper cheek heights). Greater masculinity was characterised by larger distances in all 10 features. We have now provided a more detailed description of the feature selection process and the characterisation of facial masculinity in the Method and Results sections (see lines 179-190 and lines 227-233). We have also included an extra figure with an extended caption which explains in greater detail how the facial masculinity score was calculated using the LDA space (see Figure 2 on pg. 11). This figure includes composite images derived from the current data set which illustrate the changes in features from highly feminine to highly masculine across LDA space.

4. The authors invoke genetics of facial masculinity in the discussion (and is an implicit theme of the premise), but there was not much context given or connections of this hypothesis to the broader literature on this topic, e.g., PMID: 24213680, PMID: 33288918, PMID: 34108004, and others.

Response:

We have now included the McKenna et al. (2021) paper which was published during the review of the initial manuscript (see lines 93-108), as well as other relevant works on the genetics of facial masculinity (see lines 109-115).

5. Was the LDA classifier trained only on this adult cohort?

Response:

Yes, the LDA classifier was trained only on this adult cohort. Ideally, we would have separate datasets for developing the LDA classifier and then applying it to assess facial masculinity associated with autism, as we did for our previous studies involving children (Tan et al., 2017 and 2020a). This was, however, not possible in the current study given the limited sample size. This limitation was partly overcome by using a 10-fold cross-validation technique where independent samples were used for training and testing in developing the LDA classifier. We have acknowledged this limitation of our method in the discussion section (see lines 339-343). We do not believe the central results of the study are compromised by the method we employed.

6. What was the sex breakdown of the autistic children? Do the parents of autistic males have more masculine faces than the parents of autistic females?

Response:

Of the 192 autism parents included in this study, information on the autistic children's sexes were available for 179 of them (142 autistic boys and 37 autistic girls). Based on our analyses, parents' facial masculinity score did not differ between those who had male autistic children and those who had female autistic children, $F(1, 177) = 0.37, p = .54$. This

remains the case after controlling for the effects of facial area and age, $F(1, 175) = 0.53, p = .47$. As we did not previously observe any sex differences in facial masculinity among autistic children (Tan et al., 2017) or their siblings (Tan et al., 2020a), we have not reported this analysis as we do not think that there is a strong justification for expecting a difference in the current dataset.

7. Figure 2 could be improved by having lines (and the value) of the group means.

Response:

We have now added lines and mean values to Figure 2 (now labelled as Figure 3, see pg. 13).

Reviewer 2

1. The findings raise interesting questions about possible links between ASD and factors influencing masculinity. The measure of facial masculinity is central to the report, but is described at a very abstract level, eg in terms of 20 unspecified "distances", p5, and I'm not clear on why this approach was chosen. I would ask for further information and testing before acceptance, as described below. Many of my concerns arise because there is little or maybe no investigation of the underlying masculinity differences driving the effects.

Response:

We have now added further details on how facial masculinity has been established in the current study. As mentioned in our response to Reviewer 1's comment above (see point 2), we measured 26 Euclidean distances and 26 geodesic distances from the 21 landmarks. These distances were entered into the Gradient-based Efficient Feature Selection (GEFS) algorithm to evaluate all possible combinations of the 52 facial distances and select a set of distances that maximally contributed to overall accuracy in the gender classification of the current sample. GEFS selected 10 features, of which, 4 were Euclidean (nasal width, nasal tip protrusion, nasal bridge length, and upper lip length) and six geodesic distances (upper facial height, outer canthal width, forehead width, mandible height, and left and right upper cheek heights). Greater masculinity was marked by larger distances in all 10 features. We have now provided a more detailed description of the feature selection process and the characterisation of facial masculinity in the Method and Results sections (see lines 179-190 and lines 227-233). In our response to the next comment, we explain why the current method was chosen.

2. First, I am curious why the authors do not use a more conventional GMM approach to shape analysis, an initial procrustean fit followed by a sex classifier on a training set considering the coordinates of all landmarks, and then using the trained classifier to generate a predicted sex-classification for the experimental set of parents and controls. This would seem to have important advantages: (1) it would consider the position and relative

distances of all landmarks within the shape (not eg the 20 selected by the experimenters), and (2) it would also produce a size measure for each shape. A possible (3) is that this approach is widely used. Finally (4) this might allow visualisation of the group (parent vs control) face differences.

Response:

We agree that the more conventional GMM approach to shape analysis has its merits, particularly in considering relative positions and in visualisation. The first author has used GMM shape analysis in one of her previous papers (see Tan et al., 2020b) which examined the association between autistic traits reported by non-autistic adults from the general population and facial masculinity derived from 3D images. We found it challenging to quantify the characteristics of the Principal Components (PCs) that arose from the PCA of the facial models. From our experience, these PCs are usually *qualitatively* characterised through shape manipulation and visualisation (e.g., face shapes associated with height and BMI were described based on synthetic face modelling in Holzleitner et al., 2014 but the precise facial characteristics linked to height and BMI were not reported). While such an approach is not fundamentally flawed and is commonly used in the perception literature, PCs are more difficult to replicate and generalise compared to our current method of measuring facial distances between a set of landmarks and then using GEFS and LDA. In our previous work involving facial masculinity in children, we were able to establish a set of 11 facial distances selected by the GEFS algorithm that maximally distinguished between faces of 53 non-autistic boys and 48 non-autistic girls with 98.1% gender classification accuracy (Tan et al., 2017). In a follow-up study published in Tan et al. (2020a), we measured the same set of 11 facial distances in an independent sample of 40 non-autistic boys and 40 non-autistic girls to test the generalisability of our previously established model, which generated a gender classification accuracy of 95.7%. Research prior to the present study also demonstrated the utility of the GEFS-LDA method of gender classification in adults. First, Gilani et al. (2014) reported that this approach was 99.93% accurate in gender classification, and a gender score derived from it correlated highly with perceived ratings provided by human raters for each sex. Similarly, Whitehouse et al. (2015) reported accuracy in gender classification of 99.47% using the current method.

It is also worth noting that our previous study using the GMM approach in shape analysis (Tan et al., 2020b) was a replication study of an earlier study we conducted in 2015 which investigated the relationship between autistic traits reported by non-autistic adults and facial masculinity determined by facial distance measurements and LDA classifier (see Gilani et al., 2015). Both the Tan et al. (2020b) and Gilani et al. (2015) studies generated the same pattern of findings. Therefore, from our existing evidence, we think that the method of facial analysis is unlikely to change the outcomes of the current study.

Given the existing evidence of the utility of the GEFS-LDA approach in differentiating male and female adult faces (Gilani et al., 2014; Whitehouse et al., 2015) and that our previous studies involving children with a clinical diagnosis of autism and their siblings employed

the same method for investigating facial masculinity, we think it is appropriate to use the same method in the present study to allow us to compare our observations across these datasets.

Size measures were obtained and reported in the manuscript (see lines 188-190 for measurement method and lines 235-240 for descriptive statistics). We appreciate that it might be useful to readers to visualise changes in facial structures according to the varying degrees of facial masculinity and have now included a figure depicting this (see Figure 2 on pg. 11).

3. Second, it is of course best practice to train the classifier (in whatever form) on a different sample than that used to test the classifier results. In this case (3d scans of men and women) there isn't a shortage of materials available, and this would be preferable to the 10-fold validation currently used.

Response:

We agree that it would be ideal to have separate samples for training and testing the classifier. While there are several open-source databases of 3D images available, information on the family history of autism is not usually available in these datasets. As we have observed in the current data that having a family history of autism could influence facial masculinity, we have chosen to use the 10-fold cross-validation technique where independent samples were used for training and testing. We have also acknowledged this limitation in our method in the discussion section (see lines 339-343).

4. Third, the automated landmark process used here does not seem to capture any of the facial outline, particularly the jaw. This is strange as this is the jaw and brow are key regions for human dimorphism. Shouldn't the jawline be included?

Response:

We apologise for the confusion. The 21 landmarks used in the current study were pre-specified by the authors. The automated landmarking programme merely identified the 21 pre-specified landmarks automatically which were then manually checked by the first author. We have excluded landmarks around the jaw as they are palpable landmarks that are difficult to identify from the 3D images. The identification of landmarks around the brow regions also tends to be influenced by cosmetic interventions (e.g., brow tattoos or trimming). As noted above, there is considerable evidence that the landmarks and distances we used can differentiate male and female faces with high levels of accuracy, and a facial masculinity score derived using these distances correlates highly with human ratings of masculinity.

5. Fourth, men and women seem to differ in their mouth shape at "neutral", where men present slightly less evidence of a smile than women. This might possibly drive the masculinity score, yet we don't know from the present analyses. Whether a masculinity difference is driven by a potentially controllable cue, such as expression, or by a more biologically driven one, such as brow distance, would be very interesting to know. So again I highly encourage a more thorough investigation of the shape differences driving any effect.

Response:

All participants in the current study were instructed to present a neutral facial expression during photography hence it is unlikely that differences in facial expressions between males and females may be driving the masculinity difference. However, we do acknowledge the presence of subtle facial expressions presented in supposedly neutral expressions. In the current study, we included facial distances around the lip regions (e.g., mouth width, upper and lower vermillion heights) but these distances were not selected by the GEFS algorithm as being a significant feature in distinguishing between males and females. Therefore, it is unlikely that differences in subtle smiling drove the differences observed in the current study.

6. Fifth, the confound of "facial area" and masculinity is unfortunate. I don't believe it is possible to tell from the report whether the classification algorithm is simply using some aspect of distance. For example, if parents of autistic children weighed more than the controls, and therefore had wider faces, would this be sufficient to generate a "masculinity" difference using the reported analyses?

Response:

It is possible that weight differences between the two parent groups may contribute to differences in facial distances and masculinity scores. However, the conclusions of the current study were based on analyses of the distances and masculinity scores that were conducted while controlling for effects of facial area (and age). Our analyses have shown greater facial distances and masculinity scores among the autism parents over and above any effects of facial area and age.

7. Therefore, I would suggest a more conventional GMM approach to generating a predicted masculinity score, based on a sample outside the experimental group, and which then looks backward to see where and what the masculinity differences are that drive any parent-control difference.

Response:

We have outlined above our rationale for using the GEFS-LDA approach rather than the GMM approach to shape analysis and our concerns with existing databases for training the classifier. We hope the Reviewer is satisfied with our reasoning.

Reviewer 3

1. This is a very interesting and important study showing masculinised facial features in parents of autistic children. The findings are fascinating and the methods are appropriate. In the abstract, it would be clearer if the authors could use Cohen's d instead of partial eta squared because the factors and covariates included in the ANCOVA model are not listed in the abstract.

Response:

We have now reported Cohen's d in the abstract and, for consistency, throughout the manuscript.

2. In the introduction, it would be useful if the authors could go into a bit more detail regarding the findings from the Danish biobank cohort. There was no relationship between amniotic testosterone and autism, and only males were included in the study. It would also be useful to include more recent literature showing no relationship between early androgen exposure and autistic traits (e.g., Kung et al., in press, JCPP).

Response:

We have now provided further details on the Danish biobank cohort in the introduction section (see lines 68-76), as well as more recent papers suggested by the Reviewer (see line 68).

3. As for participants, were those control adults parents as well? How was family ASD history assessed?

Response:

Yes, the control adults were parents as well. Information on family history of autism was based on self-reports. We have now described this in the Method section (see lines 153-156).

4. Regarding analyses, is it possible to include a set of analyses focusing on specific facial landmarks/features in supplementary materials?

Response:

As mentioned in our responses to comments by Reviewers 1 and 2, we have now included the specific landmarks and features analysed in the current study (see lines 167-183, Table 1 on pg. 10, lines 241-254, and Table 2 on pg. 15).

5. In the discussion, the inconsistencies regarding the links amongst early androgen exposure, autism/autistic traits, and facial masculinity need to be acknowledged. Early androgen exposure seems to relate to facial masculinity but not autism/autistic traits. I do appreciate that it is very challenging to explain and integrate these complex findings in the literature. However, at the very least, I think the inconsistencies should be acknowledged. Also, has there been any research comparing facial structures/features between autistic and neurotypical adults? Finally, are there any other confounding variables, other than age and facial area, that could have contributed to the observed group difference?

Response:

We agree that the mixed evidence related to early androgen exposure and autism/autistic traits is difficult to reconcile. Given that early androgens were not directly measured in the current study, we feel that our data is inadequate in addressing this inconsistency. We have, however, approached this discussion from a different angle by reviewing all existing studies on facial masculinity related to autism/autistic traits (see lines 306-334). In our discussion, we note that the facial distances characterising masculinity expressed in parents are somewhat different to the masculinised facial phenotype expressed in autistic children and their siblings. We speculate that this difference could be due to pubertal influences on the expression of facial masculinity. In addition, we also discuss the inconsistency in patterns of findings related to facial masculinity and autistic traits whereby females with high levels of autistic traits were found to present more masculinised facial structures whereas males with high levels of autistic traits were found to present *less* masculinised facial structures (see Gilani et al., 2015 and Tan et al., 2020). Given that parents of autistic children typically report higher levels of autistic traits, we were intrigued to find that there were no parent group \times sex interactions observed in the current study. We attribute this observation to possible differences in the biological mechanisms underlying the relationship between autistic traits and facial masculinity among those with and those without a family history of autism.

There have not been any studies that examined facial masculinity in autistic adults. We now mention in the discussion that a study of this kind would be valuable (see lines 320-324). We are unable to think of any other possible confounding factors.